# Ammonium Uptake, Mediated by Ammonium Transporters, Mitigates Manganese Toxicity in Duckweed, *Spirodela polyrhiza*

**DOI:** 10.3390/plants12010208

**Published:** 2023-01-03

**Authors:** Olena Kishchenko, Anton Stepanenko, Tatsiana Straub, Yuzhen Zhou, Benjamin Neuhäuser, Nikolai Borisjuk

**Affiliations:** 1Jiangsu Key Laboratory for Eco-Agricultural Biotechnology around Hongze Lake, Jiangsu Collaborative Innovation Centre of Regional Modern Agriculture and Environmental Protection, Huaiyin Normal University, West Changjiang Road 111, Huai’an 223000, China; 2Leibniz Institute of Plant Genetics and Crop Plant Research (IPK), 06466 Gatersleben, Germany; 3Institute of Cell Biology and Genetic Engineering, National Academy of Science of Ukraine, Acad. Zabolotnogo Str. 148, 03143 Kyiv, Ukraine; 4Institute of Crop Science, Nutritional Crop Physiology, University of Hohenheim, 70593 Stuttgart, Germany

**Keywords:** duckweed, *Spirodela polyrhiza*, manganese toxicity, ammonium transporter, gene expression, transcription factors

## Abstract

Nitrogen is an essential nutrient that affects all aspects of the growth, development and metabolic responses of plants. Here we investigated the influence of the two major sources of inorganic nitrogen, nitrate and ammonium, on the toxicity caused by excess of Mn in great duckweed, *Spirodela polyrhiza*. The revealed alleviating effect of ammonium on Mn-mediated toxicity, was complemented by detailed molecular, biochemical and evolutionary characterization of the species ammonium transporters (AMTs). Four genes encoding AMTs in *S. polyrhiza*, were classified as *SpAMT1;1*, *SpAMT1;2*, *SpAMT1;3* and *SpAMT2*. Functional testing of the expressed proteins in yeast and *Xenopus* oocytes clearly demonstrated activity of SpAMT1;1 and SpAMT1;3 in transporting ammonium. Transcripts of all *SpAMT* genes were detected in duckweed fronds grown in cultivation medium, containing a physiological or 50-fold elevated concentration of Mn at the background of nitrogen or a mixture of nitrate and ammonium. Each gene demonstrated an individual expression pattern, revealed by RT-qPCR. Revealing the mitigating effect of ammonium uptake on manganese toxicity in aquatic duckweed *S. polyrhiza*, the study presents a comprehensive analysis of the transporters involved in the uptake of ammonium, shedding a new light on the interactions between the mechanisms of heavy metal toxicity and the regulation of the plant nitrogen metabolism.

## 1. Introduction

Together with carbon, hydrogen and oxygen, nitrogen (N) is an essential nutrient that affects all aspects of the growth, development and metabolic responses of plants [1,2,3]. Nitrate (${\mathrm{NO}}_{3}^{-}$) and ammonium (${\mathrm{NH}}_{4}^{+}$) are the two major forms of N supply for plants [4]. With different plant species demonstrating preferences for either nitrate or ammonium as a preferential source of nitrogen [5,6,7,8,9], many studies revealed the dependence between nitrogen resource and the plants’ responses to stresses [10,11]. A number of studies also demonstrated that the form of supplied nitrogen affects the response of different plant species to stress caused by heavy metals such as aluminum, cadmium, copper, iron or mercury [12,13,14,15,16].

For example, the alleviating effect of ${\mathrm{NH}}_{4}^{+}\;$compared to ${\mathrm{NO}}_{3}^{-}$ related to Al [15,17] and Cd toxicity [16] has been demonstrated in rice and wheat [18]. Similarly, increasing the ratio of ${\mathrm{NH}}_{4}^{+}\;$to ${\mathrm{NO}}_{3}^{-}$ slowed down Cd absorption and reduce Cd toxicity in tomato [19]. Ammonium was demonstrated to interact with iron homeostasis in *Brachypodium* [20], and alleviate iron deficiency in rice [21]. The ${\mathrm{NO}}_{3}^{-}$/${\mathrm{NH}}_{4}^{+}\;$ration influenced copper phytoextraction causing oxidative stress, and activating the antioxidant system in *Tanzania guinea* grass [22].

Manganese (Mn) plays a crucial role as a cofactor for numerous key enzymes [23] in all living organisms and occurs naturally in soil, sediment, air borne particulates and at relatively low levels in water reservoirs. It is of special importance for plants, where it is required for proper functioning and oxygen generation by photosystem II [24]. Despite its importance, Mn is required in relatively low amounts, usually between 20 and 40 µg/kg of the crop’s dry biomass [11,23]. Deficiency of Mn may cause inhibition of growth, reduction in biomass [25] and could result in tissue necrosis due to decrease in Mn-dependent superoxide dismutase (MnSOD) and corresponding increase in free oxygen radicals [26]. On the other hand, excessive levels of Mn may be toxic for green vegetation. Plants are able to accumulate Mn in great access [27], which could lead to the Mn toxicity, usually manifested by symptoms of chlorosis, necrosis, deformation of young leaves [28,29] and characteristic brown spots on mature leaves [30]. High concentrations of this microelement can also prevent the uptake and translocation of other essential elements such as Ca, Mg and Fe inducing their deficiencies which is considered as an explanation for the inhibition of chlorophyll biosynthesis and growth [31,32]. Moreover, modern human activities such as mining, metal smelting and the application of fertilizers and fungicides to agricultural land have raised the Mn content in soil, sediments and groundwater [33,34,35,36]. These developments pollute the environment, negatively affect ecosystems and lead to accumulation of Mn in the food chain [37,38].

Duckweeds are a group of small fast growing aquatic plants, attracting increasing interest of the scientific community not only as a new model plant for various physiological, biochemical and molecular research [39], but also as a source of viable biomass [40] and a plant of choice for wastewater phytoremediation of many heavy metals [41]. Similar to rice growing on paddy fields [5,7], aquatic duckweeds have clear bias for ammonium as a source of nitrogen when given a choice between ammonium and nitrate [42]. In our previous study [43], we have demonstrated that great duckweed, *Spirodela polyrhiza*, is sensible to Mn toxicity and manifests characteristic symptoms of Mn stress when concentrations of Mn exceed 40 mg/L (0.73 mM).

In this study, we analysed the effects of 50-fold excessed Mn on various growth parameters of *S. polyrhiza* at the background of different combinations of ${\mathrm{NO}}_{3}^{-}\;$and ${\mathrm{NH}}_{4}^{+}$. The revealed physiological characteristics of biomass accumulation, N consumption and changes in cultivation medium pH, were related to comprehensive characterization of the molecular and biochemical properties of the four ammonium transporters SpAMT1;1, SpAMT1;2, SpAMT1;3 and SpAMT2, considered as the major entry pathways for ${\mathrm{NH}}_{4}^{+}$ uptake.

## 2. Results

### 2.1. Parameters of the Experimental System: Dynamics of Biomass Accumulation, Nitrogen Uptake, Changes in Medium pH, during 12-Days of Duckweed Cultivation

The effects of Mn at 2.95 mM ${\mathrm{MnSO}}_{4}$, 50-fold higher compared to the routinely used concentration in SH duckweed cultivation medium, were evaluated in dependence of two sources of inorganic nitrogen, supplied in form of nitrate (5 mM ${\mathrm{KNO}}_{3}$) or a mixture of nitrate and ammonium (2.5 mM ${\mathrm{KNO}}_{3}$ plus 2.5 mM ${\mathrm{NH}}_{4}H_{2}{\mathrm{PO}}_{4}$) during 12 days. No significant differences either in the biomass accumulation or plant appearance were observed up to four days of cultivation. However, by day 12 we observed significant, almost two-fold reduction in *S. polyrhiza* biomass accumulation in a medium supplied with nitrate and high Mn (N-Mn50) compared to nitrate medium with usual concentration of Mn (N-Mn1). Simultaneously, almost no difference in biomass accumulation was registered between duckweed grown on a mixture of nitrate and ammonium supplied with high (M-Mn50) of normal (M-Mn1) amounts of Mn (Figure 1).

Suppression of biomass accumulation in the N-Mn50 medium was accompanied by severe chlorosis and death of whole duckweed fronds profoundly manifested at day 12. To the contrary, after 12 days cultivation, duckweed grown in the medium supplied with ${\mathrm{NH}}_{4}^{+}$ (M-Mn50), mostly remained green with partial local browning of frond apexes, but no total chlorosis observed. In course of duckweed growth, the relevant changes in media such as consumption of nitrogen and pH have been monitored (Figure 1).

The dynamics of ${\mathrm{NO}}_{3}^{-}$ consumption by *S. polyrhiza* growing on N-Mn1 and N-Mn50 media were pretty similar, with the available nitrogen reduced by about a half by day 12. Duckweed cultivated in the media containing both ${\mathrm{NO}}_{3}^{-}$ and ${\mathrm{NH}}_{4}^{+}$ (M-Mn1 and M-Mn50) preferentially utilized ammonium, with nitrate almost untouched in the M-Mn50 and only slightly consumed in M-Mn1 by day 12 (Figure 1C).

During the growth period, we observed contrasting dynamics of medium acidity between medium containing ${\mathrm{NO}}_{3}^{-}$, as a soul source of nitrogen and a mixture of ${\mathrm{NO}}_{3}^{-}$ and ${\mathrm{NH}}_{4}^{+}$. The original medium pH of 5.8 climbed up to 7.01–7.18 in the medium supplied with nitrate, whereas the pH gradually dropped to 3.94–4.02 on day 12 in the medium containing both nitrate and ammonium. There were no differences in pH dynamics in the pairs of media containing usual or high concentration of Mn, N-Mn1 and N-Mn50 or M-Mn1 and M-Mn50 (Figure 1D).

### 2.2. Genome of S. polyrhiza Containes Four Genes Coding for Ammonium Transporters

A blast search on the *S. polyrhiza* genome (ecotype Sp9509 [44]), available on the NCBI website (taxid: 29656, GCA_900492545.1), using the protein sequences of rice ammonium transporters in the queries revealed four genes potentially encoding ammonium transporters SpAMT1;1, SpAMT1;2, SpAMT1;3 and SpAMT2. The exon/intron structures of the gene sequences were deduced by their similarities with the corresponding rice sequences, showing three intron-free genes belonging to the *AMT1* subfamily and only *SpAMT2* containing two short introns. A similar arrangement of the *AMT* genes has been revealed also for the second species in the genus *Spirodela*, *S. intermedia* (Appendix A) by using BLAST against the corresponding genome at the NCBI website (taxid: 51605). To validate the sequences available in the GenBank, we sequenced the amplification products for the four *SpAMTs* derived from cDNA of ecotype NB5548 used in this study (the corresponding sequence accession numbers are: OP730321, OP730322, OP730323 and OP730324).

Analysis of the deduced polypeptide sequences using InterPro software (http://www.ebi.ac.uk/interpro/ accessed on 10 September 2022) revealed structural organization typical for ammonium transmembrane transporters with characteristic N-terminal non-cytoplasmic domains of different length (53 and 50aa for SpAMT1;1 and SpAMT1;2, and 15 and 23aa for SpAMT1;3 and SpAMT2) and cytoplasmic C-terminal domains ranging from 51aa for SpAMT1;3 to 62aa for SpAMT1;2 [45] (Appendix A).

Phylogenetic analysis of the deduced protein sequences showed that SpAMT1;1 and SpAMT1;2 are closely related to each other, clustering together with their homologues from *S. intermedia* and taro, *Colocasia esculenta,* a representative of Araceae, the sister family to Lemnaceae [46]. The sequences also demonstrate significant similarity to more remote monocotyledonous species (date palm, *Phoenix dactylifera;* Cocos, rice and wheat), as well as the representative of dicotyledonous, *Arabidopsis thaliana* (Figure 2). Similar phylogenetic distances relative to other monocotyledonous and dicotyledonous were revealed also for SpAMT1;3 and SpAMT2.

### 2.3. Functional Activity of the Ammonium Transporters in Yeast and Xenopus Oocytes Systems

To test general transport function of the identified potential SpAMTs we performed a complementation analysis by expressing the four duckweed ammonium transporter proteins from the pDR199 vector in a yeast strain deficient in AMT function (31019b, ΔΔΔmep; [49]). The yeast was grown on medium supplied with ${\mathrm{NH}}_{4}^{+}$ as a sole source of Nitrogen at two concentrations, 0.3 and 1 mM and three different pH: 4.5, 5.5 and 6.5 (Figure 3). The highest complementation activity in all experimental settings, demonstrated the duckweed ammonium transporter SpAMT1;3, followed by the SpAMT1;1. The complementation efficiency of SpAMT1;3 was comparable to that of Arabidopsis AtAMT1;1, with the activity of SpAMT1;1 resembling that of AtAMT2. We were not able to detect any complementing activity of SpAMT1;2. The SpAMT2 showed some activity slightly above the pDR199 control only at 1 mM concentration of ${\mathrm{NH}}_{4}^{+}$ and medium pH 6.5. In general, there was no pH effect on any of the transporters. When grown on 60 mM methylammonia (MeA), a toxic analog of ${\mathrm{NH}}_{4}^{+}$, the active transporters conferred transport activity of MeA as well, which led to strongly reduced yeast growth.

To analyse the two functional duckweed AMTs in more detail, the proteins were expressed in *Xenopus leavis* oocytes and the ammonium induced currents where recorded by two electrode voltage clamp. Ammonium induced significant inward currents in the oocytes expressing SpAMT1;1 and SpAMT1;3 **(**Figure 4). As expected, the currents increased with more negative membrane potential (Figure 4A). Total currents by SpAMT1;3 were in the low nano ampere range and increased with increasing ammonium concentration with a low affinity K_m_ = 5.4 mM. The SpAMT1;1 mediated currents were about three times higher than the currents by SpAMT1;3 (Figure 4B) and saturated with a high affinity K_m_ = 116 µM tipical for plant AMTs (Figure 4C). No currect induction was detected in the oocytes transfected with SpAMT1;2 or SpAMT2 (data not shown). Based on the obtained results, we can characterise SpAMT1;1 as a tipical high affinity plant ammonium transporter while SpAMT1;3 showed uncommon low affinity for ammonium.

### 2.4. Expression Profiling of S. polyrhiza AMT Genes by qRT-PCR

The expression of the *SpAMT* genes was evaluated at the growth stage following 4 days of cultivation in the four media formulations mentioned above, using the primers represented in Appendix A according a previously described RT-qPCR protocol [42]. All RT-qPCR reactions were performed in triplicate, normalized against the expression of two housekeeping genes (*β-actin* and *histone H3*), and normalized to the gene expression levels at the starting point (SP). The expression of *S. polyrhiza* genes encoding AMT1;1 and AMT1;3, which demonstrated clear activity in oocytes and yeast tests, showed a complex and distinct regulation by Mn and the source of nitrogen. Adding of nitrogen in form of ${\mathrm{NO}}_{3}^{-}\;$at the background of physiological concentration of Mn (N-Mn1) led to a slight activation of both *SpAMT1;1* and *SpAMT1;3,* whereas a mixture of ${\mathrm{NO}}_{3}^{-}\;$and ${\mathrm{NH}}_{4}^{+}$ stimulated expression of *SpAMT1;3* but did not *SpAMT1;1* (Figure 5). The 50-fold elevated concentration of Mn affected the expression of *SpAMT1;1* and *SpAMT1;3* in opposite directions, enhancing *SpAMT1;1* and suppressing *SpAMT1;3*, with these effects manifested much stronger in the presence of nitrate as a sole source of N, N-Mn50, compared to nitrate and ammonium mixture, M-Mn50.

The expression of *SpAMT1;2,* was downregulated in all experimental samples compared to SP. On the contrary, *SpAMT2,* which similar to *SpAMT1;2,* did not show a function either in oocyte or in the yeast test, demonstrated a 3.3-fold increased level of transcripts in the N-Mn50 sample, while expression was downregulated both in the M-Mn1 and the M-Mn50 samples compared to the starting point.

### 2.5. Regulatory Cis-Elements in the Promoters of Spirodela AMT Genes

To get further insights into the expression regulation of *S. polyrhiza AMT* genes, we conducted an in silico analysis of cis-elements in the gene promoters. The 1.5 kb long promoter sequences upstream from the protein translation starts, extracted from the available online genome sequence, version 3 of *S. polyrhiza*, strain 9509 (GCA_900492545.1) were analyzed using the New Place promoter analysis software (https://www.dna.affrc.go.jp/PLACE, accessed on 7 October 2022) [50]. In parallel, we conducted a similar analysis of corresponding gene promoters in a recently published genome of *S. intermedia* (taxid: 51605) a closely related duckweed species, with the aim to further highlight the evolutionary conserved regulatory elements (Figure 6).

A prominent structural feature revealed in all analyzed promoters, was the abundant presence of GAGA motifs or its complement CTCT, involved in regulation of eukaryotic genomes through chromatin modulations [51,52,53]. Especially intriguing is the presence of a more than 50 bp long GAGA stretch in promoters of *AMT1;1* gene of *S. polyrhiza* and *S. intermedia*, closely adjacent to the ATG of protein translation start. Within the first 500 bp upstream of the translation start, all analyzed promoters feature one (*AMT1;1* and *AMT1;3*) or multiple (*AMT1;2* and *AMT2*) TATA motifs. The TATA-box is recognized as binding sites for a basic transcription factor TBP (TATA box-binding protein) involved in transcription initiation of many eukaryotic genes [54]. Moreover, inside 500 bp proximity to the ATG, all *AMT1* genes promoters contain general eukaryotic *cis-*element CCAAT-box [55] a target for transcription complex NF-Y (Nuclear Factor Y), which plays an essential role in many processes related to plant development, productivity and stress responses [56,57]. The *SpAMT2* and *SiAMT2* promoters have multiple CCAAT sites located between −1000 and −500 bp relating to the first ATG.

Transcription factors IDD (Indeterminate Domain) and DOF (DNA binding with one finger) were shown to be involved in regulation of the *AMT* genes in a number of different plants [58]. We located the recognition site for IDD, with the signature ACAAA, within the first 500 bp at the identical positions in the *AMT2* promoters of both *S. polyrhiza* and *S. intermedia.* In the promoters of the *AMT1* genes the site is scattered at various locations more remote from the translation start. As for the DNA motif recognized by DOF, T/AAAAG, multiple copies are found within the first 500 bp of all *AMT1* promoters, as well as scattered at more distant locations in promoters of both *AMT1* and *AMT2* genes.

The members of the NIGT1 (Nitrate-Inducible GARP-Type Transcriptional Repressor-1) subfamily in the superfamily of GARP transcription factors [59], participate in fine-tuning of gene expression in response to the plant nitrogen and phosphorus status [60]. The proteins bind to two types of *cis*-elements, GAATC and GAATATTC demonstrating dual modes of promoter recognition [61,62]. The GAATATTC site has been identified only in the *AMT2* promoters of *S. polyrhiza* and *S. intermedia* about 550 bp upstream of the first ATG, while the GAATC recognition site was located along the sequence of all promoters with the highest representation in the promoters of *AMT1;2* and *AMT2* genes.

## 3. Discussion

Our previous study demonstrated that the extend on Mn toxicity in four aquatic species of duckweed (Lemnaceae) depends on the source and concentration of nitrogen in the cultivation medium [43]. In particular, in great duckweed, *Spirodela polyrhiza*, the first characteristic symptoms of Mn stress appeared in form of characteristic brown spots when concentration of Mn was 40 mg/L (0.73 mM); however, the symptoms were largely reduced by the presence of ammonium in the medium in addition to nitrate. This observation, together with (i) increasing pollution of water resources with heavy metals [34,35,36,37], (ii) multiple previous reports linking the level of toxicity caused by various heavy metals to the nitrogen metabolism [12,13,14,15,16] and (iii) availability of the solid set of data on whole *Spirodela* genome sequences [45,63,64], encouraged us to take a closer look at Mn toxicity in *S. polyrhiza* in relation to the source of inorganic nitrogen with a special attention on the species’s ammonium transporters (AMTs).

### 3.1. $NH_{4}^{+}$ Alleviated Mn Stress Symptoms for S. polyrhiza Compared with $NO_{3}^{-}$

Monitoring of *S. polyrhiza* fronds cultivated in the medium containing either a basic (0.059 mM) or 50-fold elevated (2.95 mM) concentration of Mn revealed profoundly different growth outcome, depending on the nitrogen source. Duckweed grown in the medium with basic Mn supplemented with ${\mathrm{NO}}_{3}^{-}$, as a sole nitrogen source (N-Mn1), looked perfectly healthy and produced higher biomass compared to the medium with a mixture of a 50% of ${\mathrm{NO}}_{3}^{-}$ and 50% of ${\mathrm{NH}}_{4}^{+}$ (Figure 1A). This observation is in accordance with the previously published data showing that growth rate of two duckweed species, *S. polyrrhiza* and *Lemna aequinoctialis* was lower in the medium containing ammonia compared to the medium with nitrate [65]. In addition to reduced biomass, selected fronds in the M-Mn1 medium showed some signs of chlorosis. These symptoms can be attributed to general ammonium toxicity [29,30,31], which we also observed in our previous study on six duckweed species, including *S. polyrhiza* [42].

The picture dramatically changed when the cultivation medium was supplied with elevated Mn. The fronds’ appearance and biomass accumulation remained practically the same for duckweed in the medium with mixture of ${\mathrm{NO}}_{3}^{-}$^-^ and ${\mathrm{NH}}_{4}^{+}$ (M-Mn50) compared to M-Mn1, except of occasional brown spots characteristic for Mn-mediated stress. In contrast, duckweed cultivated in the N-Mn50 medium with ${\mathrm{NO}}_{3}^{-}$ as a sole N source, manifests severe senescence, probably even death, and significant reduction in biomass accumulation compared not only to the N-Mn1 medium, but also to duckweed grown in medium containing the mixture of ${\mathrm{NO}}_{3}^{-}$ and ${\mathrm{NH}}_{4}^{+}$ supplied with either normal (M-Mn1) or high concentration of Mn (M-Mn50). It is worth to notice that high concentration of Mn had no effect on the dynamics of ${\mathrm{NO}}_{3}^{-}$ and ${\mathrm{NH}}_{4}^{+}$ uptake or the dynamics of pH during the cultivation period (Figure 1C,D**)**. Confirming our previous study [42], utilization of ${\mathrm{NO}}_{3}^{-}$ led to a pH increase in the medium, whereas the uptake of ${\mathrm{NH}}_{4}^{+}$ resulted in acidification of the nutrient medium, as has been shown earlier for a number of plants [42,66,67].

The lowering of pH in the media containing ${\mathrm{NH}}_{4}^{+}$ might be a part of the mechanism of the observed alleviation of Mn toxicity mediated by ${\mathrm{NH}}_{4}^{+}$ in *S. polyrhiza*. For example, ${\mathrm{NH}}_{4}^{+}$ was shown to alleviate Mn stress symptoms in spruce and rice [68,69], and the decrease in Mn uptake by medium acidification was proposed as a mechanism underlying this phenomenon [69]. pH sensitivity has been reported as a general feature of transmembrane solute movement [70] as well as for many kinds of specific transporters in different organisms [71,72,73,74]. The reported data demonstrate that conformation and functionality of the plant transmembrane transporters are dependent upon cellular and intercellular pH. Dependence on pH, is especially well established in relation to the activity of aquaporins [75,76]. Aquaporins represent a large family of plant proteins with members in the model plant *Arabidopsis thaliana* [77] responsible not only for the plants water homeostasis, but for transporting many other molecules including ammonia and other small solutes [78]. Amounting evidence from across phyla suggests that some aquaporin types act as ion channels, and are modulated by divalent cations such as ${\mathrm{Ca}}^{2+}$ and ${\mathrm{Cd}}^{2+}$ [79]). Moreover, in *Arabidopsis,* in addition to pH, manganese was identified as potent inhibitors of aquaporin AtPIP2;1 [80].

In order to check the possibility that aquaporins could contribute to Mn stress in our system by responding to Mn and/or ammonium/pH, we have tested expression of two duckweed homologues of *Arabidopsis* aquaporins, *SpTIP2;1* and *SpPIP2;2* by quantitative RT-qPCR (Figure 7, Appendix A). Indeed, both genes showed differential expression in response to the applied experimental conditions. The *SpTIP2;1* was significantly suppressed in the N-Mn50 environment but slightly upregulated in the M-Mn1 and M-Mn50, whereas *SpPIP2;2* was upregulated in all experimental medium, especially in the medium containing ${\mathrm{NH}}_{4}^{+}\;$(M-Mn1 and M-Mn50).

We can assume that the acidification of the cultivation media and probably Mn itself may inactivate the transporters supporting inward Mn transport into the cytosol and lead to Mn exclusion from plant tissues. The presented data on differential expression of aquaporins may pave a way for future research in this area.

### 3.2. Ammonium Transporter (AMT) Genes in the Genome of S. polyrhiza

Ammonium transporters (AMTs) are transmembrane proteins, usually consisting of a N-terminal non-cytoplasmic domain, 11 transmembrane helices and C-terminal cytoplasmic domain [81,82], responsible for transporting ${\mathrm{NH}}_{4}^{+}\;$from the outside into the inside of the cell. Intensive studies on AMTs from different plant species revealed that this protein family is divided into two distinct subfamilies, the AMT1 and the AMT2, represented by different number of members in different species [58,83]. Compared to other plants, *S. polyrhiza* contains fewer genes coding for the AMTs. For example, *Arabidopsis thaliana* has five genes coding for AMT1 variants and one gene for AMT2 [83], and rice has three members of the AMT1 subfamily [84] and nine representatives of AMT2 [85]. Our analysis of the *S. polyrhiza* genome revealed three variants of AMT1: SpAMT1;1, SpAMT1;2, SpAMT1;3 and a single SpAMT2, all encoded by single copy genes located on different chromosomes. This finding further illustrates the reduced number of protein-coding genes in *S. polyrhiza* originally observed by the first genome sequencing [63], where a 28% reduction in genes compared to *Arabidopsis thaliana* and 50% reduction compared to rice was shown. This was later confirmed by Michael et al. (2017) [45] showing that most gene families in *Spirodela* have less members as compared with the model dicotyledonous plant *Arabidopsis* and monocotyledonous *Brachypodium*. Sequence analysis of the cloned cDNA from the Chinese ecotype NB5548 shows very high homology to the corresponding sequences of European ecotype 9509 (BlastN GCA_900492545.1) at the level of 99.6–99.8% similarity, with just a few observed C↔T or G↔A substitutes in the 1500 bp long sequences of the three *SpAMT1* genes, and 100% similarity for *SpAMT2*. This level of sequence conservation between ecotypes of *S. polyrhiza* is in a good agreement with the previously postulated extremely low rate of mutations revealed by comparing genomes of 68 worldwide geographic ecotypes of this species [86]. The *SpAMT* gene sequences also show similar arrangement, no introns in all three AMT1 gene sequences, two short introns within the AMT2 gene, and 95–97% sequences similarities with their homologues in the genome of the closely related *S. intermedia* [64], as represented in Appendix A.

The deduced SpAMT protein sequences demonstrate most close phylogenetic relationship to their counterparts from *S. intermedia*, followed by taro, *Colocasia esculenta* [87], a plant species which belongs to the group of early-branching monocotyledonous family Araceae [88], a sister family to Lemnaceae [46]. The SpAMT proteins also demonstrate significant levels of similarity to the homologues from other monocotyledonous banana, cocos, oil palm, rice and wheat, as well as dicotyledonous *Arabidopsis* (Figure 2), stressing strong evolutionary conservation of this important protein family along the plants [83,89]. Among *S. polyrhiza* AMTs, SpAMT1;1 and SpAMT1;2 share the highest homology, whereas the two proteins have certain differences in their C-termini (Appendix A). This offers possibilities for differential post-translational regulation since the proteins’ C-terminal domain was reported as a target for environmental regulation through phosphorylation by protein kinases [82,90,91,92].

### 3.3. SpAMT1;1 and SpAMT1;3 Demonstrate Clear Activity in Yeast and Oocytes

From the four SpAMT proteins expressed in yeast and *Xenopus* oocytes, SpAMT1;1 and SpAMT1;3 demonstrated highly specific activity in parallel with the *Arabidopsis* controls (Figure 3). Interestingly, the homologues of these transporters in *Arabidopsis*, AtAMT1;1 and AtAMT1;3, have been reported for conferring 60% of total ammonium uptake combined [93,94]. To the contrary, SpAMT1;2 showed no activity in all of the experimental conditions and SpAMT2 was slightly activated at elevated pH in yeast complementation test. One might ponder about the silence of the SpAMT1;2, considering its phylogenetic closeness with the SpAMT1;1 (Figure 2) which shows a quite good activity in transporting ${\mathrm{NH}}_{4}^{+}$. However, when looking on the structure of both proteins in more details, one can see that despite a high homology of the transmembrane part in the middle of the proteins, the non-cytoplasmic N-termini domains and the cytoplasmic C-termini domains show significant differences (Appendix A). For example, specific mutations of a conserved Gly in this domain was shown to inactivate AMTs from yeast and plants [49,95]. The mentioned mutation, however, cannot explain the functional differences between SpAMT1;1 and SpAMT1;2. The Gly is intact within the conserved first half of the C-terminal domain in both proteins. Another well-documented option of AMT regulation, is by post-translational phosphorylation of the Thr, located 4 residues downstream of the above mentioned Gly [96]. This position also does not differ between SpAMT1;1 and SpAMT1;2, as evident from the protein alignment in Appendix A. However, additional Thr residues uniquely positioned within both N-terminal and C-terminal domains of SpAMT1;2 (Appendix A), offer an opportunity of differential regulation of this protein compared to SpAMT1;1, through phosphorylation. These suggestions are purely speculative at the moment, and require an experimental verification.

### 3.4. The Four S. polyrhiza AMT Genes Show Different Expression Patterns in Response to Nitrogen Source and Mn

Our data show that both nitrogen source and elevated concentration of Mn affect expression of each of the *SpAMT* genes in a unique and in some instances opposite direction under the applied experimental conditions (Figure 3). Although ${\mathrm{NO}}_{3}^{-}$ slightly stimulated expression of *SpAMT1;1* and *SpAMT1;3,* it downregulated *SpAMT1;2* and *SpAMT2.*
${\mathrm{NH}}_{4}^{+}$ was rather neutral on *SpAMT1;1,* upregulated *SpAMT1;3* and downregulated *SpAMT1;2* and *SpAMT2.* The revealed wide spectrum of *SpAMT* gene expression patterns in response to nitrogen is not too surprising. It is well in line with multiple previous reports showing diversity of AMT responses in other species such as *Arabidopsis* [97], rice [98], maize [99], poplar [14,100], apple [101] or tobacco [102]. The availability of multiple *AMT* genes with individual expression patterns, and multiple capacity of the expressed transporters for ${\mathrm{NH}}_{4}^{+}$ uptake is considered to provide plants with great flexibility to respond differentially to varying environmental and nutritional conditions by fine tuning gene expression.

Simultaneous, presence of high concentrations of Mn in the cultivation media containing nitrate, N-Mn50, or a mixture of nitrate and ammonium, M-Mn50, dramatically changes the expression of *SpAMT* genes compared to the media with physiological concentration of Mn, N-Mn1 and M-Mn1. Thus, we observed drastic increase in expression of *SpAMT1;1* and *SpAMT2* in the medium containing ${\mathrm{NO}}_{3}^{-}$ as a sole source of nitrogen, N-Mn50, whereas the increase in the M-Mn50 medium was rather mild (Figure 5). Based on this specific expression dynamics, we are inclined to explain the observed expression boost not by direct influence of Mn, but rather by the process of senescence started by Mn toxicity and manifested in fronds cultivated in the N-Mn50 to a significantly greater extend, compared to the duckweed grown in the M-Mn50 (Figure 1). Observations in other plants, showing upregulation of the *SpAMT1;1* and *SpAMT2* homologues in old tissues, suggesting the role of these proteins in nitrogen remobilization [103], might be used in support of this notion. For example, van der Graaff et al., (2006) [104] revealed enhanced expression of *AtAMT1.1* in senescing leaves of *Arabidopsis*, and more recently, Liu et al., (2018) [102] showed that of nine AMT transporter genes identified in tobacco, only two, *NtAMT1.1* and *NtAMT2.1*, were strongly expressed in the old leaves, implying their dominant roles in N remobilization from senescent tissues.

Contrary to *SpAMT1;1* and *SpAMT2*, expression of the *SpAMT1;2* and *SpAMT1.3* seems to be influenced by Mn in a more direct manner. High Mn concentration suppressed expression of both genes independent of the: (i) supplied nitrogen formulation, N-Mn50 or M-Mn50 as, respectively, compared with N-Mn1 or M-Mn1 (Figure 5) and (ii) the differences in the acidity of the cultivation medium, correspondingly pH~6.5 for N-Mn1 and N-Mn50 and pH~ 4.0 for M-Mn1 and M-Mn50 (Figure 1). The question whether expression of *SpAMT1;2* and *SpAMT1.3* is directly controlled by Mn or through some Mn-mediated physiological changes such as hormonal status, generation of reactive oxygen species (ROS), or some other signaling pathway remains to be investigated.

### 3.5. Distinctive Promoter Elements in the AMT Genes in S. polyrhiza

Many environmental and physiological factors, nitrogen status, soil pH, photoperiod, developmental stage [58], have been recognized to influence ammonium transporter expression in plants. These factors regulate expression of *AMT* genes primarily via interaction of various transcription factors with specific *cis*-elements in the promoter of the genes [55]. Our analysis of *AMT* gene promoters in *Spirodela*, revealed characteristic *cis*-element patterns which might affect transcriptional regulation of ammonium uptake, and are distinctive from the genes encoding major nitrogen assimilation genes responsible to both ${\mathrm{NO}}_{3}^{-}$ and ${\mathrm{NH}}_{4}^{+}$ [42].

The promoters of *AMT* genes are populated by (GA/CT)_n_ repeats (Figure 6), targets for GAGA/TCTC-binding transcription factors (GAFs). Abundance of the GAGA in the promoters is not very surprising, considering that genomes of both *S. polyrhiza* and *S. intermedia*, are un-proportionally enriched in these motifs [45,64]. However, the contrast in distribution of the sequences in *AMTs* and nitrogen assimilation genes promoters is noticeable. In the analyzed *AMT* promoters, relatively short GAGA-stretches are more or less evenly distributed along the 1.5 kb length of the promoter, with a unique ~50 bp GAGA-stretch located just upstream of the ATG in the *AMT1;1* promoter (Figure 6), whereas in most of the promoters of nitrogen assimilation genes the (GA/CT)_n_ repeats are represented by long stretches located outside of the 500 bp region adjacent to the genes’ translation start.

Among prominent conservative *cis*-elements present within the 500 bp region upstream of translation start in all analyzed *Spirodela AMTs* are the TATA-box, a binding site for the general eukaryotic transcription factor TBP [54]. All *AMT* promoters also contain another general eukaryotic *cis-*element CCAAT-box [55], a binding site for NF-Y (nuclear factor Y) [105], involved in many important growth and developmental processes [106], including stress and hormone responses [107,108].

Overall, the range of *cis*-elements revealed in the *Sprirodela AMT* genes promoters in combination with the wide selection of corresponding transcription factors provide huge flexibility to regulate the *AMT* transcripts in response to various developmental, environmental and nutritional challenges faced by a plant during the life span.

## 4. Summary

Here we showed that the toxicity of manganese in duckweed plants could be mitigated by addition of ammonium as a nitrogen source. This mitigation effect might be pH related since the preferential uptake of ammonium by *S. polyrhiza* resulted in a pH decrease in the media which in turn might lead to a decreased activity of Mn uptake transporters e.g., aquaporins. The uptake of ammonium from the media could be mediated by four SpAMT proteins. The expression of the *AMT* genes was uniquely regulated by Mn and the available nitrogen source and two of the AMT proteins showed strong activity in heterologous systems. Their interesting unique transcriptional regulation is orchestrated by multiple transcription factor binding sites in the *AMT* promoters. The exact mechanism of Mn resistance by ammonium as well as the identification of transcription factors involved in *SpAMT* regulation are interesting questions which should be addressed in future studies.

## 5. Materials and Methods

### 5.1. Plant Material

The *S. polyrhiza* ecotype (collection ID: NB5548) used in this study was selected from the duckweed live in vitro collection recently established in the School of Life Sciences at Huaiyin Normal University, Huai’an, China. The NB5548 ecotype was originally collected in fall 2017 (N 33″17′40; E 118″49′45), surface sterilized and propagate on solid agar medium supplemented with SH salts [109] under sterile conditions.

### 5.2. Duckweed Cultivation Parameters

To accumulate biomass, duckweed plants grown on solid agar medium were initially transferred into 200 mL sterile liquid SH medium supplemented with 5 g/l sucrose contained in 500-mL flasks and cultivated at 23 ± 1 °C with a photon flux density of 50–60 μmol·m^−2^s^−1^ provided by cool white fluorescent bulbs in a 16-h light/8-h dark cycle. After four weeks of growth, the accumulated fronds were washed 3 times with autoclaved water, and 200 mL sterile N- and sugar-free basal salt SH medium was added. After 3 days cultivation, 0.5-g portions were blotted and weighed in aseptic condition and then inoculated into 100-mL flasks containing 60 mL of autoclaved basic SH medium (no sugar) supplemented with two different formulations of N (N and M) and two concentrations of ${\mathrm{MnSO}}_{4}$ (0.059 mM or 2.95 mM), namely N-Mn1, N-Mn50, M-Mn1 and M-Mn50. The basic medium labeled Mn1 contained 0.059 mM Mn that correspondent to SH medium formula, whereas experimental medium Mn50 contained 50-fold elevated concentration of Mn (2.95 mM). The media N-Mn1 and N-Mn50 contained ${\mathrm{NO}}_{3}^{-}\;$as the sole N source (5 mM ${\mathrm{KNO}}_{3}$, with 5 mM ${\mathrm{KH}}_{2}{\mathrm{PO}}_{4}$ replacing ${\mathrm{NH}}_{4}H_{2}{\mathrm{PO}}_{4}$ of the standard SH medium), and the media M-Mn1 and M-Mn50 had a mixture of ${\mathrm{NO}}_{3}^{-}$ and ${\mathrm{NH}}_{4}^{+}$ (2.5 mM ${\mathrm{KNO}}_{3}$, 2.5 mM ${\mathrm{KH}}_{2}{\mathrm{PO}}_{4}$, 2.5 mM ${\mathrm{NH}}_{4}H_{2}{\mathrm{PO}}_{4}$ and 2.5 mM $K_{2}{\mathrm{SO}}_{4}$). The medium pH was adjusted to 5.5 and autoclaved. In total, 18 flasks with growing duckweed have been prepared for every variant of tested media. Three flasks (as 3 independent biological repeats) were randomly taken after 4 and 12 days for each of the analyses (RNA isolation for RT-qPCR, biomass measurement, determination of the total concentration of N). The duckweed biomass was also collected at the starting point (designated as SP), which corresponds to the moment before washing and 3-days cultivation on a medium containing no nitrogen (starvation).

### 5.3. Determination of N Concentration

The total nitrogen concentration was determined using standard alkaline potassium persulfate digestion followed by UV spectrophotometry as previously described [110]. The ${\mathrm{NO}}_{3}^{-}$ concentration in the growth media was measured spectrophotometrically as the difference in absorption between 220 and 275 nm [111]. The ${\mathrm{NH}}_{4}^{+}$ concentration in the growth media was measured calorimetrically using the Nessler method [112].

### 5.4. Cloning and Analysis of S. polyrhiza Genes Encoding Ammonium Transporters

The *AMT1;1*, *AMT1;2*, *AMT1;3* and *AMT2* genes were obtained by sequencing and cloning the PCR-amplified gene regions using cDNA prepared from local *S. polyrhiza*, ecotype NB5548, mRNA as a template. The PCR fragments were amplified with gene-specific primers designed according to the in silico sequence information available at NCBI (taxid: 29656, GCA_900492545.1) for *S. polyrhiza*, ecotype 9509. The generated DNA fragments, cloned into the vector pMD19 (Takara, Dalian, China) following the manufacturer’s instructions were custom sequenced (Sangon Biotech, Shanghai, China), and the obtained nucleotide sequences were analyzed using the CLC Main Workbench (Version 6.9.2, Qiagen) software. The specific primers used for gene amplification are listed in Appendix A.

### 5.5. Phylogenetic Analysis

Protein sequences of AMT proteins were compared with a selection sequence available in public databases. The corresponding sequences available for monocotyledonous and dicotyledonous were extracted from the GenBank (National Center for Biotechnology Information (NCBI). Available online: https://www.ncbi.nlm.nih.gov (accessed on 15 August 2022) by blasting with the *S. polyrhiza* sequences as a query. The maximum-likelihood phylogenetic trees were constructed using NGPhylogeny webservice accessible through the https://ngphylogeny.fr [47] using MAFFT Multiple Sequence Alignment [113] and PhyML algorithm [114] with the Jones–Taylor–Thornton (JTT) model. Cleaning aligned sequences was made by utilizing BMGE tools [115]. Bootstrap support (BS) was estimated with 100 bootstrap replicates. iTOL (https://itol.embl.de) was used for displaying and annotating the generated phylogenetic trees [48].

### 5.6. Construction of Expression Vectors and Testing AMTs Functionality in Yeast and Xenopus oocytes

For testing in yeast and oocytes the coding sequences of *SpAMP1;1, SpAMP1;2, SpAMP1;3 and SpAMP2* were cut out from the original plasmid pMD19 and ligated into the vector pDR199 for expression in yeast [82] or the oocyte expression vector pOO2 [116]. Following verification by sequencing, the resulting vectors pDR199-SpAMP1;1, pDR199-SpAMP1;2, pDR199-SpAMP1;3 and pDR199-SpAMP2 were introduced into the AMT-deficient yeast strain 31019b [49]. The complementation analysis was performed according to the earlier described procedure [81], using the pDR199 vector as negative control and the pDR199 featuring *Arabidopsis thaliana* genes *AtAMT1;2* and *AtAMT2* as positive controls. Correspondingly, the four *S. polyrhiza AMT* genes were fitted into the pOO2 vector, and the resulting expression cassettes were used to test the proteins’ abilities for transporting ${\mathrm{NH}}_{4}^{+}$ in the oocyte according to an earlier established protocol [92]. Oocytes were ordered at EcoCyte Bioscience (Castrop-Rauxel, Germany), selected, washed and injected with 50 nl of linearized cRNA (0.4 µg/µL) of *SpAMTs*. Oocytes were kept in ND96 for 3 days at 18 °C and then placed in a recording chamber containing the recording solution (in mM): 110 CholineCl, 2 ${\mathrm{CaCl}}_{2}$, ${\mathrm{MgCl}}_{2}$, 5 N-morpholinoethane sulfonate (MES), pH adjusted to 5.5 with Tris(hydroxymethyl) aminomethane (TRIS). Ammonium was added as Cl salts. Oocytes were impaled with 3 M KCl-filled glass capillaries of around 0.8 Mohm resistance connected to a two-electrode voltage clamp amplifier (Dagan CA-1). Transport was measured at an array of ${\mathrm{NH}}_{4}^{+}$ concentrations (10, 30, 100, 300, 1000, 3000 and 10,000 µM). Currents without ammonium were subtracted from currents with ammonium to give ammonium induced current.

### 5.7. Analysis of AMT Genes Expression in S. polyrhiza by RT-qPCR

After DNAase treatment, 600 ng of total RNA was reverse transcribed using Reverse Transcriptase cDNA synthesis kit (Takara, Beijing, China) following the manufacturer’s manual. The qPCR reactions were performed using CFX Connect Real-Time detection system (Bio-Rad, Hercules, CA, USA) using the UltraSybr Mixture (High Rox) supplied by CWBio (Taizhou, China). The cycling conditions were as follows: initial denaturation at 95 °C for 10 min followed by 40 cycles of 30 s at 94 °C, and 20 s at the annealing temperature of the respective primers (Appendix A). The SYBR Green I fluorescence was monitored consecutively after the annealing step. The quality of products was checked by a thermal denaturation cycle. Only results providing a single peak were considered. The coefficient amplification efficiency for each pair of primers was determined by 10-fold serial dilutions. The level of relative expression was calculated by the 2^−ΔΔCt^ method [117]. Expression data for the target genes were normalized using the average expression of two *S. polyrhiza* ecotype NB5548 housekeeping genes, *histone H3* (GenBank ID: MZ605911) and *β-actin* (GenBank ID: MZ605912), according to the geNorm protocol [118]. Three replicates were performed for all samples. Normalized expression was estimated using the ΔΔCq algorithm All data were analyzed using the program BIO-RAD CFX Manager 3.1 (Bio-Rad, USA) and Microsoft Excel 2016 software.

## Figures and Tables

**Figure 1 plants-12-00208-f001:**
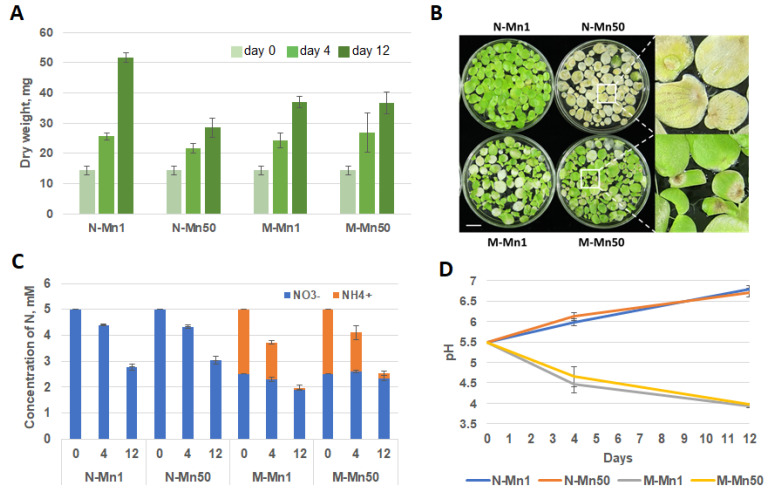
Dynamic parameters related to duckweed cultivation in media with different formulations of N and Mn: (**A**) Duckweed biomass at the beginning of the experiment, (light green) and following 4 days (green) and 12 days (dark green) cultivation in media supplied with different concentrations of nitrate, ammonium and Mn. N-Mn1—medium containing 5 mM ${\mathrm{NO}}_{3}^{-}$ and 0.059 mM Mn; N-Mn50—medium containing 5 mM ${\mathrm{NO}}_{3}^{-}$ and 2.95 mM Mn; M-Mn1—medium containing a mixture of 2.5 mM ${\mathrm{NO}}_{3}^{-},$ 2.5 mM ${\mathrm{NH}}_{4}^{+}$ and 0.059 mM Mn; M-Mn50—medium containing a mixture of 2.5 mM ${\mathrm{NO}}_{3}^{-},$ 2.5 mM ${\mathrm{NH}}_{4}^{+}$ and 2.95 mM Mn. (**B**) Images of *S. polyrhiza* fronds taken at day 12 following cultivation in media supplied with different concentrations of nitrate, ammonium and Mn; (**C**) Comparative dynamics of nitrogen uptake, supplied in the medium in form of nitrate (${\mathrm{NO}}_{3}^{-}$) or ammonium (${\mathrm{NH}}_{4}^{+}$); (**D**) Comparative dynamics of pH of four cultivation media in course of duckweed growth. Bar is 1 cm.

**Figure 2 plants-12-00208-f002:**
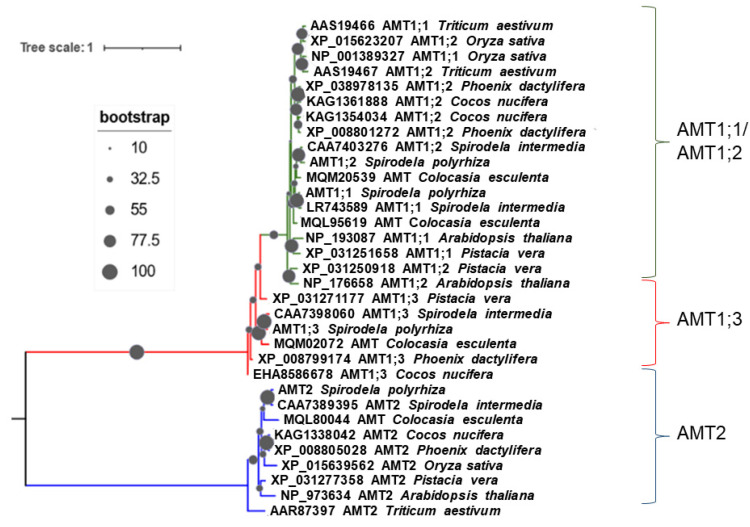
Clustering of AMT protein sequences from selected plant species. The phylograms shows a maximum likelihood tree obtained using available AMT protein sequences from different plants. Multiple alignment and phylogenetic tree were generated in NGPhylogeny webservice (https://ngphylogeny.fr, accessed on 27 September 2022) [47] and iTOL (https://itol.embl.de, accessed on 29 September 2022) [48].

**Figure 3 plants-12-00208-f003:**
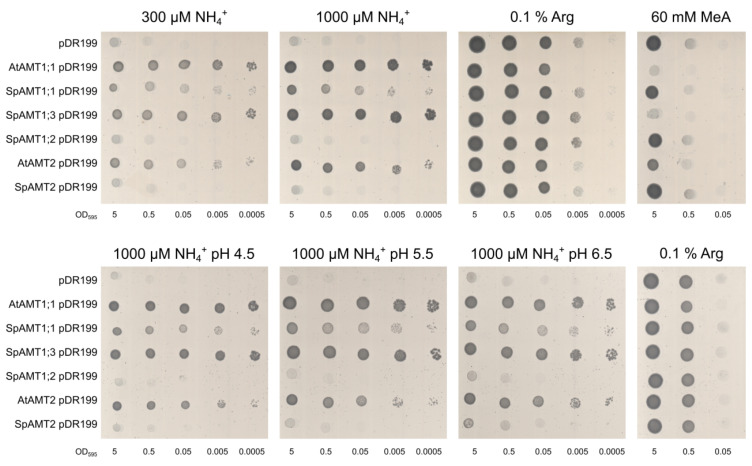
Transport activity of SpAMTs in yeast. Top panel represents growth on 0.3 or 1 mM NH_4_Cl as sole nitrogen source, 0.1 % (*w*/*v*) Arginine was used as a positive control. Growth on 60 mM MeA was used as a toxicity test, with 0.1 % (*w*/*v*) Arginine as nitrogen source. The bottom panel shows growth on 1 mM NH4Cl when media was buffered at pH 4.5, 5.5 or 6.5 unbuffered media with 0.1 % (*w*/*v*) Arginine was used as dilution control. Spotted were five 10 times dilutions starting with OD_595_ = 5. Pictures are representatives of four independent repetitions.

**Figure 4 plants-12-00208-f004:**
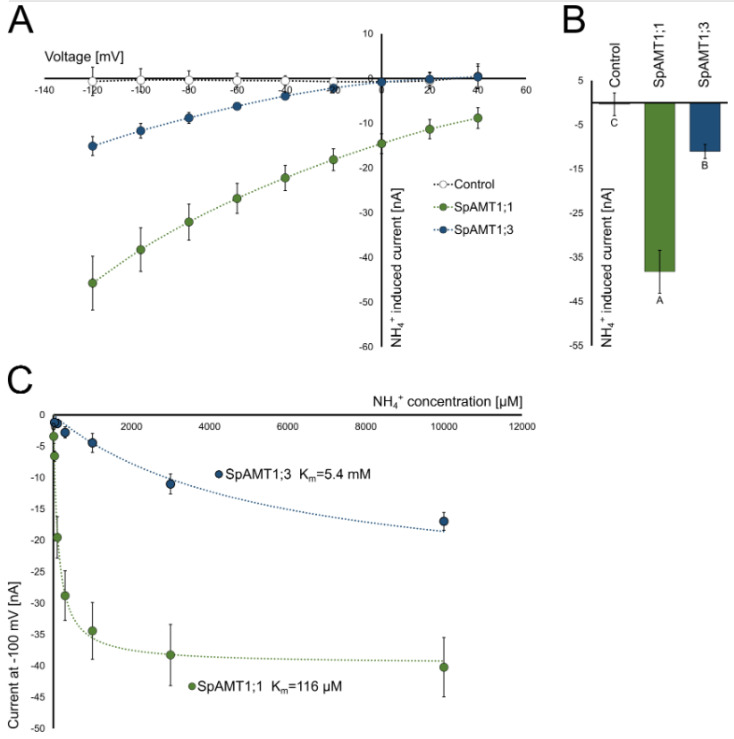
Functional characterization of SpAMT1;1 and SpAMT1;3 in *Xenopus laevis* oocytes: (**A**) Current/Voltage plot of ammonium induced currents in SpAMT1;1 and SpAMT1;3 expressing oocytes. Currents without ammonium were subtracted from currents with 3 mM ammonium; (**B**) For comparison of current intensities, ammonium induced (wash subtracted) currents by 3 mM and 100 mV are given as a bar chart; (**C**) Saturation kinetics of SpAMT1;1 and SpAMT1;3 mediated ammonium transport in oocytes using concentrations of ammonium from 0–10 mM ${\mathrm{NH}}_{4}\mathrm{Cl}$. Data show means +/− SEM; *n* ≥ 8.

**Figure 5 plants-12-00208-f005:**
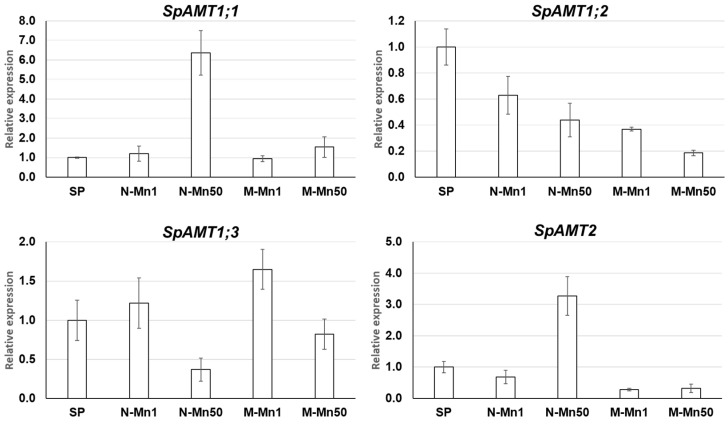
Expression patterns of the four *SpAMT* genes in duckweed cultivated in different media (N-Mn1, M-Mn1, N-Mn50 and M-Mn50). The gene expression was determined at cultivation day 4 by RT-qPCR relative to day 0, starting point (SP). Gene expression levels are in relative units. Error bars show ± SD of 3 replicates (*p* < 0.05).

**Figure 6 plants-12-00208-f006:**
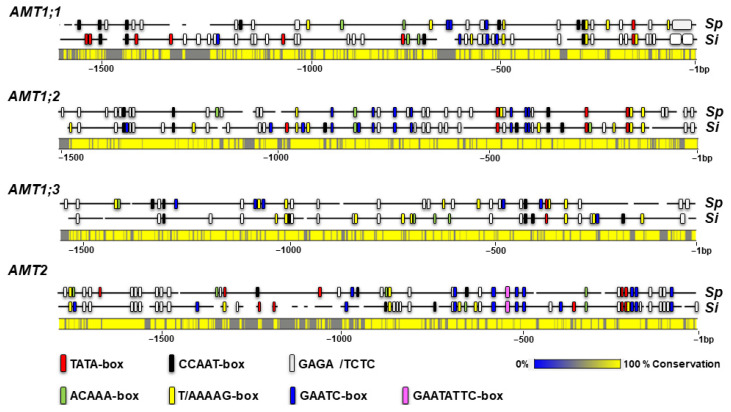
Schematic representation of promoters of the four *AMT* genes (*AMT1;1*, *AMT1;2*, *AMT1;3 and AMT2)* in the genomes of *Spirodela polyrhiza* (*Sp)* and *Spirodela intermedia* (*Si*). The negative numbers represent distance from the first ATG codon in nucleotides (bp). Colored boxes indicate locations of the corresponding promoter *cis*-elements. The conservation bar refers to the level of similarity between the homologous promoter sequences of *Spirodela polyrhiza* (*Sp*) and *Spirodela intermedia* (*Si)*.

**Figure 7 plants-12-00208-f007:**
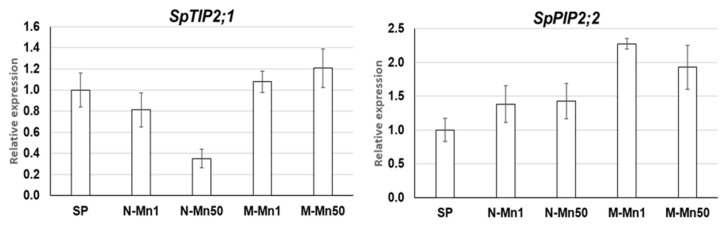
Expression patterns of two aquaporin genes, *SpTIP2;1* and *SpPIP2;2*, in duckweed cultivated in different media (N-Mn1, M-Mn1, N-Mn50 and M-Mn50). The genes expression was estimated at the cultivation day 4 by RT-qPCR relating to day 0, starting point (SP). Gene expression levels are in relative units. Error bars show ± SD of 3 replicates (*p* < 0.05).

## Data Availability

GenBank accession numbers for *Spirodela polyrhiza* AMT genes are: *SpAMT1;1*—OP730321; *SpAMT1;2*—OP730322; *SpAMT1;3*—OP730323; *SpAMT2*—OP730324.

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
