# Peer review of "Ammonium Uptake, Mediated by Ammonium Transporters, Mitigates Manganese Toxicity in Duckweed, Spirodela polyrhiza"

_plants, 2023, doi:10.3390/plants12010208_

Round 1

Reviewer 1 Report

Nitrogen is an essential nutrient that affects growth, development and metabolic response in plant. The author applied duckweed to investigate the influence of nitrate and ammonium on the Mn toxicity. The results are promising, I suggest accepting after minor revision.

1. Figure 1A, Figure 5 and Figure 7 need significant analysis.

2. Figure 6 is not clear.

3. Figure 2 species names need italics.

Author Response

Open Review 1.

Nitrogen is an essential nutrient that affects growth, development and metabolic response in plant. The author applied duckweed to investigate the influence of nitrate and ammonium on the Mn toxicity. The results are promising, I suggest accepting after minor revision.

Thank you very much for this positive assessment of our work. We hope that by the following changes we could successfully address the points raised by you thereby make our manuscript suitable for publication. 

  1. Figure 1A, Figure 5 and Figure 7 need significant analysis.

Thanks for the remarks. Explanation has been extended for the description of Figure 1A, hoping it will make the data better understandable. Regarding Figures 5 and 7, the authors follow the common practice accepted in the field of presenting and describing the RT-qPCR results. 

  1. Figure 6 is not clear.

Thanks for noting. More detailed description of the presented data has been introduced.

  1. Figure 2 species names need italics.

The species names are corrected.

Reviewer 2 Report

Thank you for having submitted a clearly structured manuscript.

In fact, your experiments were focussing on ammonia transporters rather than the biochemistry of Mn toxicity or the compartmentation of Mn and N, respectively. Therefore, please consider whether the title of your manuscript is well chosen. For instance, you are showing similar rates of N intake in all experiments while the increase in dry weight is significantly higher in the N-Mn1 experiment only. Since corresponding measurements are lacking, such obvious and maybe surprising results should be discussed on the basis of literature citations. Can you rule out that the fast growing culture fertilised by nitrate is low in polyamine content, for instance? Though plants are said to be autonomous with this respect, the Schenk & Hildebrandt medium is rich in vitamins. Is there any information available (i) on the role of vitamins in stress response, and (ii) on pH effects on vitamin intake?

Numbering of the chapters in material & methods is irritating: §5 is followed by §4.1, etc.

Author Response

Open Review 2

Comments and Suggestions for Authors

Thank you for having submitted a clearly structured manuscript.

In fact, your experiments were focussing on ammonia transporters rather than the biochemistry of Mn toxicity or the compartmentation of Mn and N, respectively. Therefore, please consider whether the title of your manuscript is well chosen. For instance, you are showing similar rates of N intake in all experiments while the increase in dry weight is significantly higher in the N-Mn1 experiment only. Since corresponding measurements are lacking, such obvious and maybe surprising results should be discussed on the basis of literature citations. Can you rule out that the fast growing culture fertilised by nitrate is low in polyamine content, for instance? Though plants are said to be autonomous with this respect, the Schenk & Hildebrandt medium is rich in vitamins. Is there any information available (i) on the role of vitamins in stress response, and (ii) on pH effects on vitamin intake?

We thank the reviewer for the careful reading of the manuscript, thoughtful notices and constructive recommendations.

Following the reviewer’s suggestion, we have modified the title, to one with a sharper focus on ammonium transporters. The new title reads as: “Ammonium uptake mediated by ammonium transporters mitigates manganese toxicity in duckweed, Spirodela polyrhiza”. 

Regarding the accumulation of dry biomass, we highlighted the difference in biomass accumulation produced by duckweed grown on N-Mn1, M-Mn1 media in the Discussion section and introduced an additional reference [65]. Concerning, the relation of N intake and the dry weight accumulation, we do not have a reasonable explanation at the moment; the only thing we can add is that in those experiments, duckweeds were cultivated using “sugar-free basal SH medium” without vitamins, as stated in the Materials and Methods.

Numbering of the chapters in material & methods is irritating: §5 is followed by §4.1, etc.

Thanks for noticing. Corrected.